# Separating Knowledge and Perception with Procedural Data

**Adrián Rodríguez-Muñoz** [1]   **Manel Baradad** [1]   **Phillip Isola** [1]   **Antonio Torralba** [1]

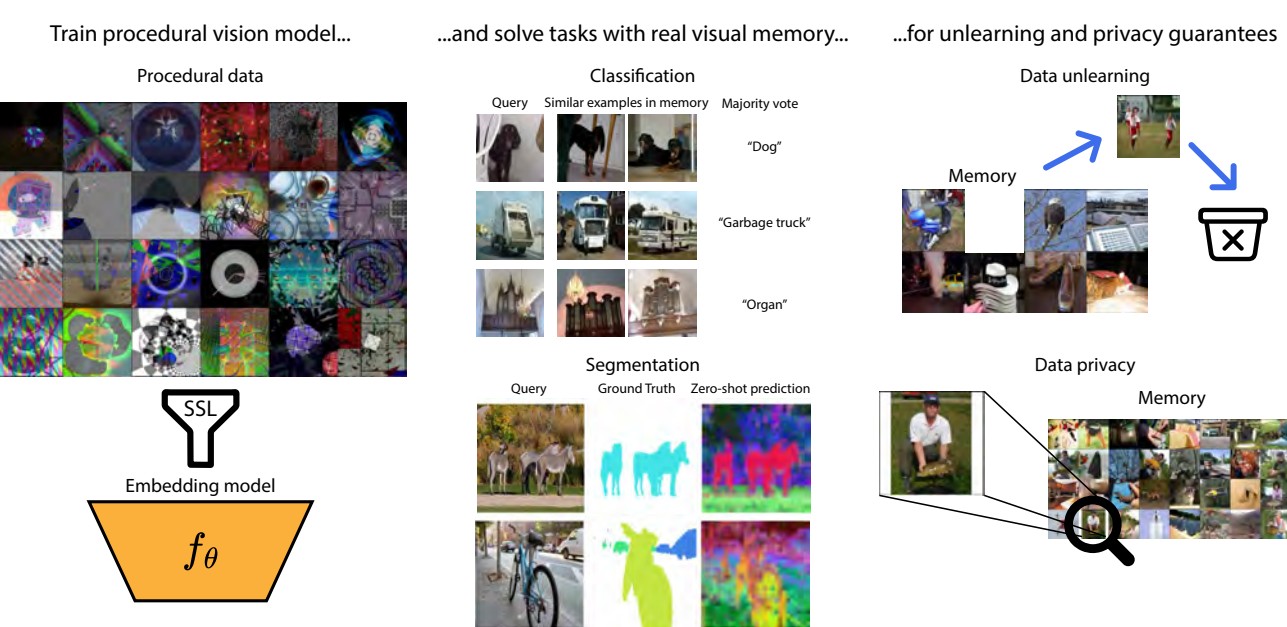

**Figure 1:** Our approach is as follows: first, an embedding model is trained on procedural data generated with OpenGL code using self-supervised learning (SSL). In this stage, unlearning and attribution is difficult, but procedural data is much less exposed to privacy/bias risk and real-world semantics. Next, we use the embedding model on real world tasks using only a visual memory of reference image embeddings, without extra training. When working with real instead of procedural data, there is high privacy/bias risk and real world semantics. However, isolating all real data to only the memory makes efficient data unlearning and privacy analysis possible. The overall system has perfect control over all real world data, while approximating the performance of standard training.

## Abstract

We train representation models with procedural data only, and apply them on visual similarity, classification, and semantic segmentation tasks without further training by using visual memory—an explicit database of reference image embeddings. Unlike prior work on visual memory, our approach achieves full compartmentalization with respect to all real-world images while retaining strong performance. Compared to a model trained on Places, our procedural model performs within 1% on NIGHTS visual similarity, outperforms by 8% and 15% on CUB200 and Flowers102

fine-grained classification, and is within 10% on ImageNet-1K classification. It also demonstrates strong zero-shot segmentation, achieving an $R^2$ on COCO within 10% of the models trained on real data. Finally, we analyze procedural versus real data models, showing that parts of the same object have dissimilar representations in procedural models, resulting in incorrect searches in memory and explaining the remaining performance gap.

## 1. Introduction

Modern vision systems learn by digesting images into weights via gradient descent. While offering strong performance, this approach carries concerns about interpretability, privacy, and bias. Moreover, it makes adding and removing data difficult, since weights store knowledge in a black-box manner. To alleviate this issue, prior work proposed the

[1]Department of Electrical Engineering and Computer Science, Massachusetts Institute of Technology, Cambridge, USA. Correspondence to: Adrian Rodríguez-Muñoz <adrianrm@mit.edu>.

*Proceedings of the 42nd International Conference on Machine Learning*, Vancouver, Canada. PMLR 267, 2025. Copyright 2025 by the author(s).

Procedural data | Realistic data

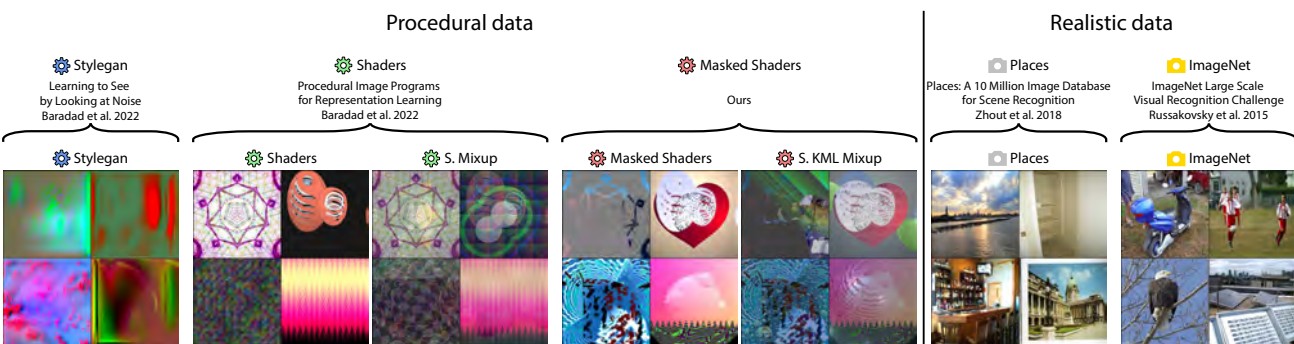

**Figure 2:** Examples of procedural data from prior work, our new Masked Shaders: Shaders KML and Shaders KML Mixup, and the real datasets Places and ImageNet. Masked Shaders have higher diversity and consistently beat prior processes in downstream tasks.

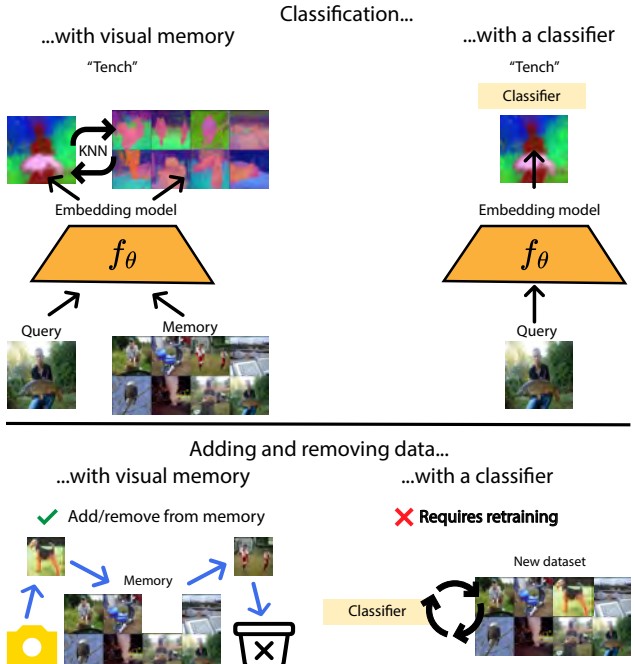

**Figure 3:** Classification with visual memory uses k-nearest neighbours (KNN) w.r.t. a database of embeddings rather than a parametric classifier (top). This allows efficient adding and removal of data through modification of the database (bottom).

idea of explicitly separating how knowledge is represented, the feature embeddings, from what knowledge is stored, the visual memory (Geirhos et al., 2024; Weston et al., 2015; Chen et al., 2018; Iscen et al., 2022; 2024; Gui et al., 2024; Silva et al., 2024). Depicted in Figure 3, they called this approach **"perception with visual memory"**.

Unlike a traditional network which feeds the query embedding through a classifier to obtain the output, perception with visual memory works by taking the query embedding, retrieving its k-nearest neigbhours (KNN) from the memory, and outputting the majority label. This in turn makes adding and removing knowledge easy and efficient: while the classifier would need to be fine-tuned or retrained, the visual memory can simply add and drop data samples. There

remains just one problem, the feature embeddings are also a parametric model trained on data. While adding and removing training samples from the memory is easy, doing so from the feature embeddings is not. In this work, we propose training the embedding model with procedural data. Unlike real world data, procedural data is non-realistic and is generated via simple code, and thus is much less exposed to the privacy or bias risks that motivate unlearning. Figure 2 compares examples of procedural and real images. Prior work focused on combining procedural embeddings with linear classifiers trained on real data (Baradad et al., 2021; 2022), and only very briefly mentioned using a neighbors approach. In this work, we delve deeper into this ability and explicitly make the connection to visual memory perception.

Our contributions are as follows:

1. We demonstrate that procedural embeddings with visual memory allow perfect unlearning and privacy guarantees w.r.t. *all* real data, while retaining strong performance.

2. We introduce the new procedural data processes Shaders KML and Shaders KML Mixup, which yield stronger embeddings than those of prior work.

3. We show that procedural embeddings possess remarkable zero-shot and in-context semantic segmentation abilities, on the same order of magnitude as embeddings trained on real data.

## 2. Related work

**Visual memory** marries the compartmentalization and interpretability of databases with the effectiveness of neural methods. This is done by applying k-nearest neighbours algorithms on trained neural embedding databases (Papernot & McDaniel, 2018). Prior work has proposed employing this technique for few-shot learning (Wang et al., 2019; Yang et al., 2020; Bari et al., 2021), adversarial robustness (Sitawarin & Wagner, 2019; Papernot & McDaniel, 2018;

Rajani et al., 2020), medical image classification (Zhuang et al., 2020), confidence calibration (Papernot & McDaniel, 2018), interpretability (Papernot & McDaniel, 2018; Rajani et al., 2020; Wallace et al., 2018; Lee et al., 2020), image denoising (Plötz & Roth, 2018), retrieval- augmented learning (Khandelwal et al., 2020; Drozdov et al., 2022), anomaly and out-of-distribution detection (Bergman et al., 2020; Sun et al., 2022), and language models (Wu et al., 2022; Khandelwal et al., 2020; Min et al., 2024). In our work, we specifically build on top of (Nakata et al., 2022) and (Geirhos et al., 2024), which showed the effectiveness of visual memory at large scale. Different from them, we use procedural data to train the embeddings, thus making *all* real data used be under database-like control.

**Procedural data** approaches train networks on non-realistic data generated via code. The benefits of this are several fold: it makes the process more interpretable and optimizable (Sun et al., 2021), can be used to improve the privacy-utility tradeoff of differentially private gradient descent (SGD) (Abadi et al., 2016; Tang et al., 2023; Choi et al., 2024), and provides insights on the human visual system. Prior work has explored using fractals (Kataoka et al., 2020; Anderson & Farrell, 2022; Nakashima et al., 2022), untrained generative networks (Baradad et al., 2021), and recently, openGL programs (Baradad et al., 2022). We contribute two new processes that obtain higher performance, and show the semantic segmentation ability of procedural models.

# 3. Train procedural embeddings

We train an embedding model using *procedural* data, similarly to (Baradad et al., 2021; 2022). Unlike typical synthetic data which approximates the target distribution using a complex generative model, procedural data is created with simple programs. See Figure 2 for the procedural and real datasets considered in this work. We train vision transformers (ViT) (Dosovitskiy et al., 2021) using the local-to-global similarity objective of DINO (Caron et al., 2021). On real data, this objective teaches models to have similar representations for parts of objects existing in reality, leaking biases and personal identities in the process. In contrast, the same objective but with procedural data teaches to have similar representations for parts of abstract shapes and textures, with much less risk of biases and privacy leakage. Procedural embeddings lack knowledge of real world entities, yet are surprisingly strong.

We evaluate the procedural networks on a Human Visual Similarity (HVS) task using the NIGHTS dataset (Fu et al., 2023), an analysis missing in prior work. This benchmark consists of a Two Alternative Forced Choice (2AFC) on trios of images: given a reference and two options, which option has greater embedding cosine similarity with the reference? The test measures ability to match human judgments on

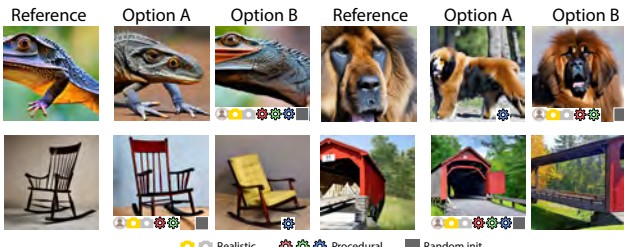

**Figure 4:** Examples from the NIGHTS dataset, along with human, realistic model, and procedural model judgments.

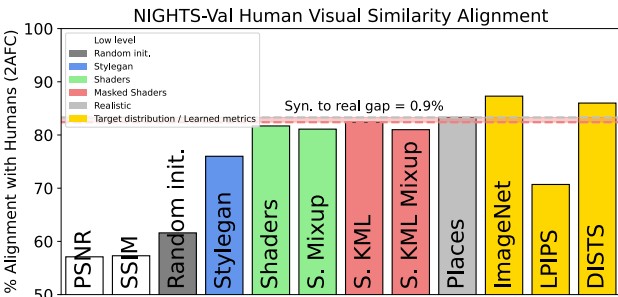

**Figure 5:** Models performance on the NIGHTS-Val benchmark. The best procedural model, trained on Shaders KML, has a high % alignment with humans of 82.4%, within 0.9% of the Places model, trained on realistic data without domain overlap.

not only low-level colors and textures, which are easily captured by simple white-box metrics, but also on mid-level similarities in layout, pose, and content, that are harder to define. Figure 4 shows examples from the dataset with human and model answers. Results in Figure 5 show that procedural data metrics have performance within 1% of the Places model, trained on real data without domain overlap. The ImageNet model has class overlap with NIGHTS, and thus is only for reference. White-box metrics like PSNR and SSIM barely perform above chance.

## 3.1. The Shaders KML Mixup process

The prior best procedural dataset is Shaders Mixup from (Baradad et al., 2022). They observed that models trained on the raw Shaders dataset learned short-cut solutions with poor generalization, but interpolating multiple samples in pixel space with Mixup (Zhang et al., 2018) alleviated the issue and achieved a new state-of-the-art.

In our work, we derive a stronger approach that extracts mixing masks from the Shaders images, rather than always

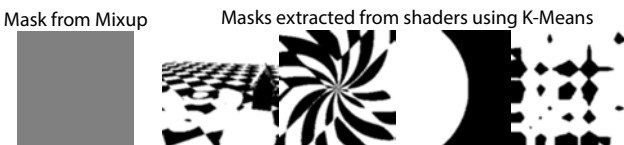

**Figure 6:** Constant mixing mask used by mixup (left) vs data driven mixing masks obtained using KMeans (right). Using the latter leads to much greater diversity.

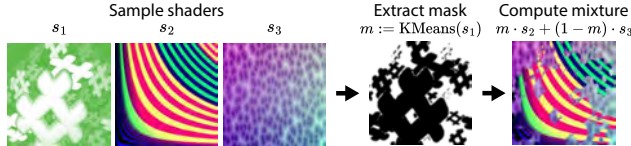

**Figure 7:** Diagram of the Shaders KML process. First, three shaders $s_1$, $s_2$, and $s_3$ are sampled. Next, $s_1$ is used to obtain a mask $m$ using KMeans in RGB space. Finally, we mix $s_2$ and $s_3$ using $m$ to obtain the Shaders KML sample.

using a constant mixing mask like in Mixup. As seen in Figure 6 this has the effect of increasing dataset diversity, found in (Baradad et al., 2021) to be one of the biggest drivers of performance for non-realistic data. We call this process, shown visually in Figure 7, Shaders K-Means Leaves (S. KML). Shaders KML obtains comparable performance to Shaders Mixup. Applying Mixup on Shaders KML to suppress short-cut solutions yields the Shaders KML Mixup process, obtaining a new state-of-the-art as seen in Table 1.

# 4. Classification and segmentation with visual memory

We apply the procedural models on classification and segmentation tasks without further training using visual memory. We compare procedural data with training on Places, a dataset of natural images different to the evaluation distribution, which acts as an upper bound to procedural data. Results for training on ImageNet, which is either the target dataset or has high overlap, is included only for reference.

**Classification** is qualitatively more challenging than the similarity task from Section 3: going from mid-level concepts in a small three image set, to higher-level semantics in a large and diverse look-up pool of up to $\mathcal{O}(1M)$ images. However, procedural models still obtain strong performance. Table 1 shows KNN classification accuracy on various fine-grained datasets and ImageNet-1K (Russakovsky et al., 2015). Remarkably, the best procedural model actually beats the model trained on Places (Zhou et al., 2018) on the fine-grained classification datasets, by 15%, 8%, and 1% on Flowers102 (Nilsback & Zisserman, 2008), CUB200 (Wah et al., 2011), and Food101 (Bossard et al., 2014) respectively. We posit that this is due to the little or no semantic overlap between Places and the three fine-grained datasets, which means the Places model spends capacity on semantic associations that are not useful. In contrast, the procedural models learn foundational skills from abstract shapes and textures that may better generalize. As a sanity check, all procedural models are worse than the ImageNet model which has semantic overlap with all three fine-grained datasets. On ImageNet-1K classification, the best procedural model is within <10% of the Places model. Figure 8 shows a query image and its nearest neighbors (NNs) for three datasets, visually showing the abilities of procedural models.

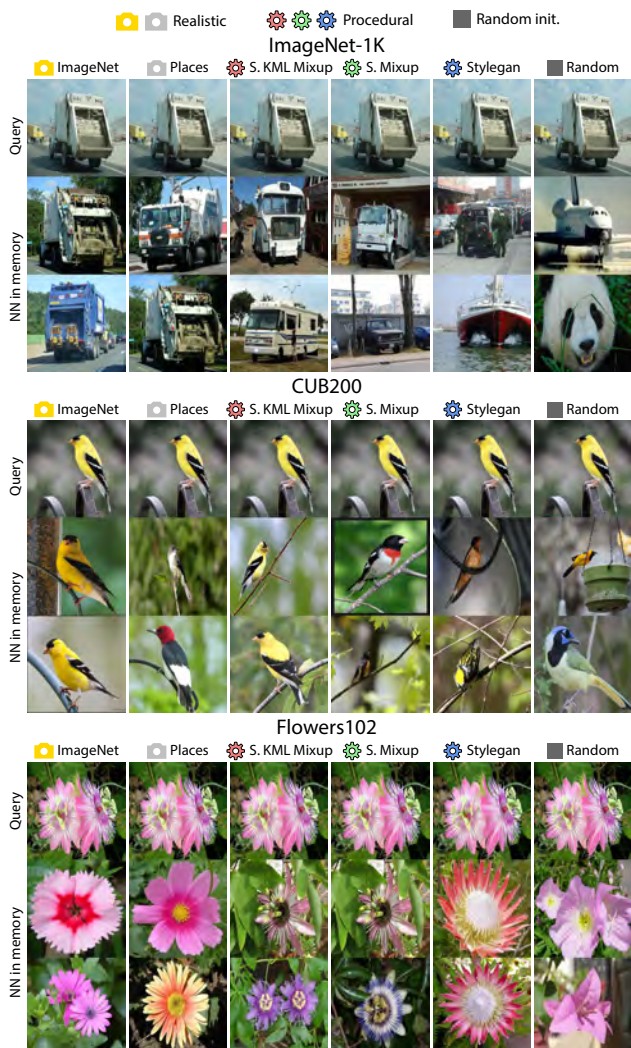

**Figure 8:** Visual comparison of query images from various datasets and their neareast neighbours according to each of the models. Procedural models can effectively search for perceptually similar images on a wide variety of datasets, despite not seeing real-world data during training.

| Data type | Dataset | Fine-grained | | | General |
| --- | --- | --- | --- | --- | --- |
| | | Flowers | CUB | Food | ImageNet-1K |
| Target | ImageNet | 83.43 | 55.20 | 64.45 | 68.89 |
| Realistic | Places | 59.51 | 19.09 | 47.78 | 47.30 |
| Procedural | S. KML Mixup | **75.20** | **27.08** | **48.70** | **37.88** |
| | S. KML | 71.86 | 24.02 | 46.00 | 35.38 |
| | S. Mixup | 73.33 | 19.85 | 47.78 | 35.45 |
| | Shaders | 66.18 | 15.59 | 38.05 | 30.69 |
| | Stylegan | 41.86 | 8.61 | 22.73 | 13.73 |
| White-box | Random init. | 11.18 | 1.93 | 5.32 | 1.84 |
| | SIFT | - | - | - | 3.08 |

**Table 1:** Performance on KNN classification with visual memory. The best procedural model, trained on Shaders KML Mixup, beats the realistic Places model on Flowers, CUB, and Food fine-grained classification, and has a gap of only 10% on ImageNet-1K.

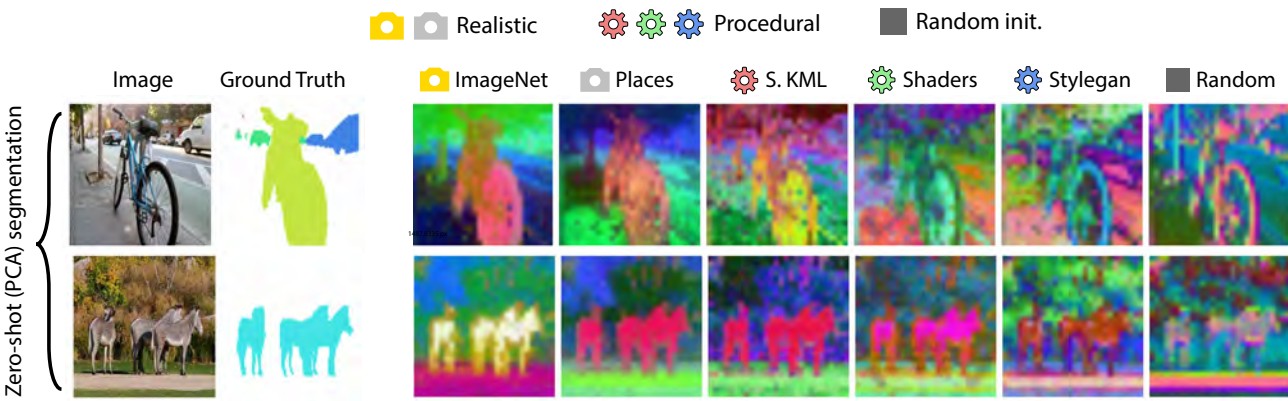

**Figure 9:** Zero-shot segmentation on COCO (Lin et al., 2015) using principal component analysis (PCA) features. Procedural models clearly separate the bike and zebra from the background. However, visually distinct parts of the bike, such as the center and spokes of the wheel, have similar and dissimilar representations in real and procedural models respectively. This is due to procedural models not having seen bikes before, while real models learn they are parts of the same object.

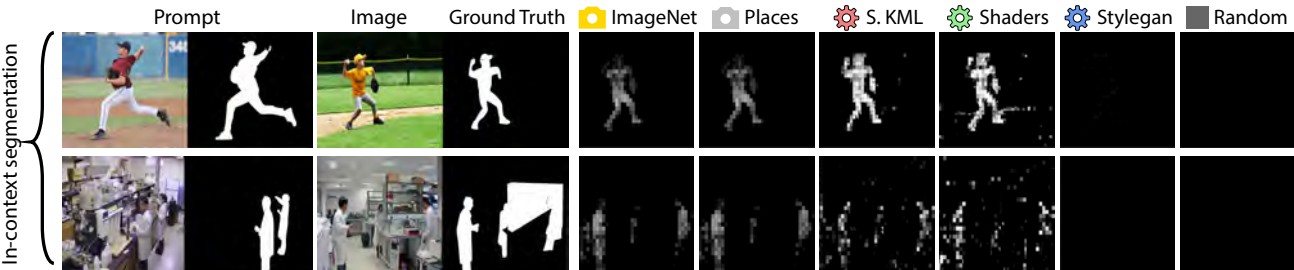

**Figure 10:** In-context segmentation on Ade20k (Zhou et al., 2017). Procedural models can segment arbitrary classes given a single exemplar prompt. This holds even in the second image, where there are many distractors of the same color.

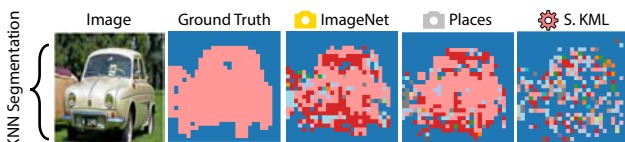

**Figure 11:** KNN segmentation on Pascal (Everingham et al., 2010). Procedural models are limited at segmenting with KNN. Due to not seeing real-world objects during training, representations of individual parts can be very dissimilar. This leads to spurious similarities with object parts of other classes, harming performance.

**Segmentation:** Procedural models have remarkable semantic segmentation ability. Figure 9 qualitatively shows how procedural features clearly separate the bike and zebras from their surroundings. Quantitatively, Table 2 shows numerical $R^2$ (ratio of explained variance to total variance) between principal component analysis (PCA) features and human labels. The best procedural model is within 10% of real data models and highly above random and RGB features. Procedural models are also capable of in-context segmentation: given a prompt image and a prompt mask representing a concept, they can effectively search for it in a new query image, even in the presence of equally colored distractions like in the second row of Figure 10. However, they struggle at KNN semantic segmentation with a large visual memory, as seen in Figure 11. As explained in Section 3, the DINO

| Data type | Dataset | COCO |
|---|---|---|
| Realistic | ImageNet | 63.7 |
| | Places | 62.1 |
| Procedural | S. KML Mixup | 53.7 |
| | S. KML | **55.9** |
| | S. Mixup | 51.4 |
| | Shaders | 55.0 |
| | Stylegan | 48.5 |
| White-box | Random init. | 36.7 |
| | RGB | 19.4 |

**Table 2:** $R^2$ of PCA features and human label segmentations.

objective on real data teaches models to have similar representations for parts of real world objects, even when the parts are visually dissimilar. In contrast, procedural models, having never seen the object during training, will have dissimilar representations for the parts. We can observe this in Figure 9: the center and spokes of the wheel are colored the same in real models and differently in procedural models. Procedural models have excessively local representations which are vulnerable to spurious similarities with object parts of other classes.

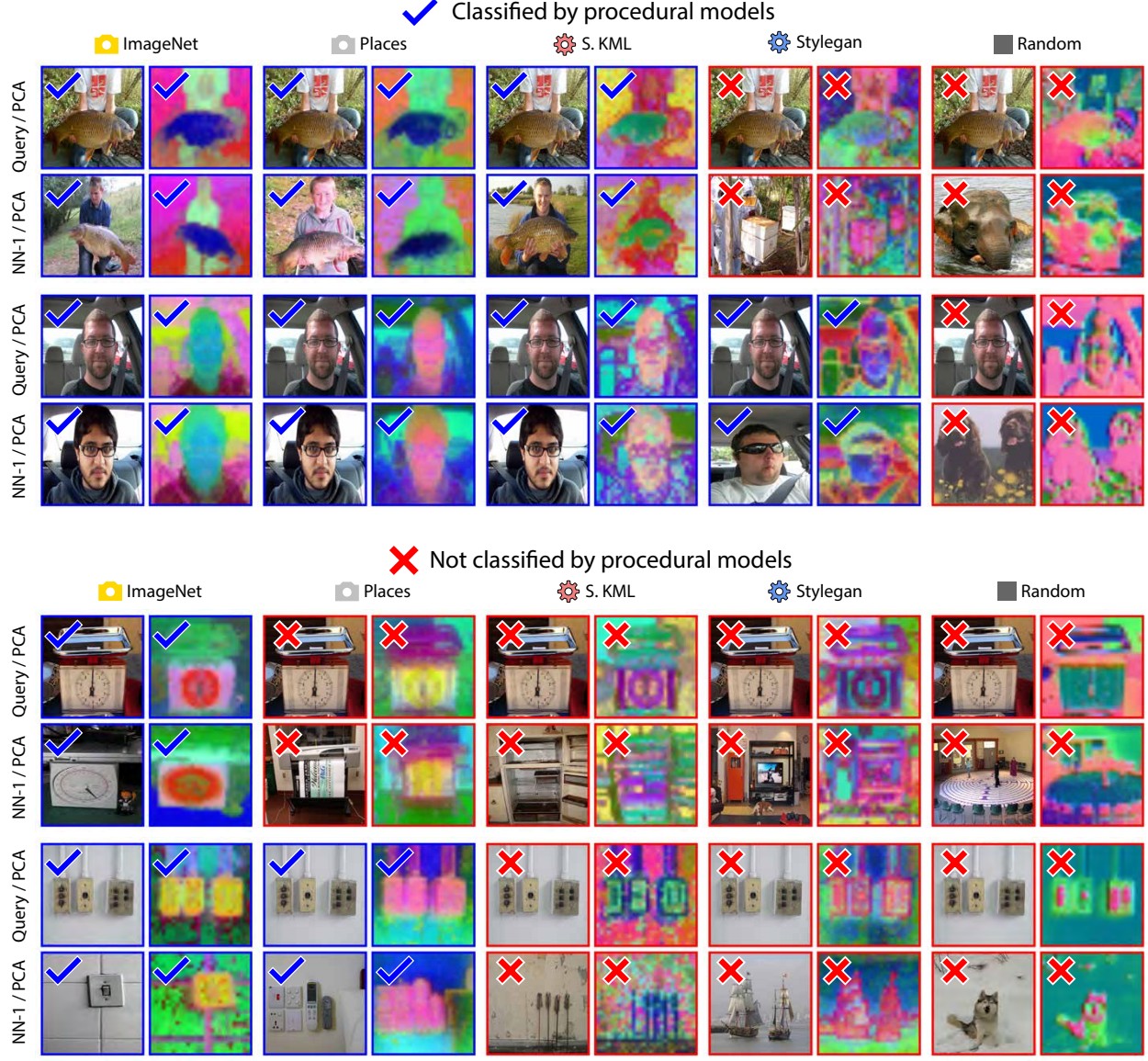

**Figure 12:** Feature PCAs of images correctly (top) and incorrectly (bottom) classified by the Shaders KML Mixup model. PCAs for correctly classified images separate distinct objects and join parts of the same object (fish, face). In the other hand, in incorrectly classified images they fail to separate distinct objects (wall and body of the plugs) or fail to join parts of the same object (hand and face of the dial).

## 5. Analysis of limitations

Despite their good performance, procedural models still lag behind models trained on real data. In this section, we develop insights on why and how this gap occurs. Figure 12 shows feature PCAs of images and their nearest neighbor in memory according to each model. We observe that correctly classified images have feature PCAs close to the natural segmentation: distinct objects are clearly separated, and parts of the same object such as the fish and the face share a single distinct color. In contrast, for images classified incorrectly PCAs fail to separate distinct objects (the switch cable and casing have very similar colors) or fail to join

parts of the same object (the hand and face of the dial have different colors), problems which are not the case in the feature embeddings trained on ImageNet.

Due to never having seen them during training, procedural models do not know to identify the object that defines the class (e.g. dial, switch) as a single entity, leading to incorrect nearest neighbours. For example, in the balance image (first incorrect example), the S. KML Mixup model separates the metallic balance, dial face, and dial hand in green, yellow, and purple respectively. The nearest neighbour *correctly* has similar looking regions, but is the *wrong* class. We leave finding ways of addressing these limitations to future work.

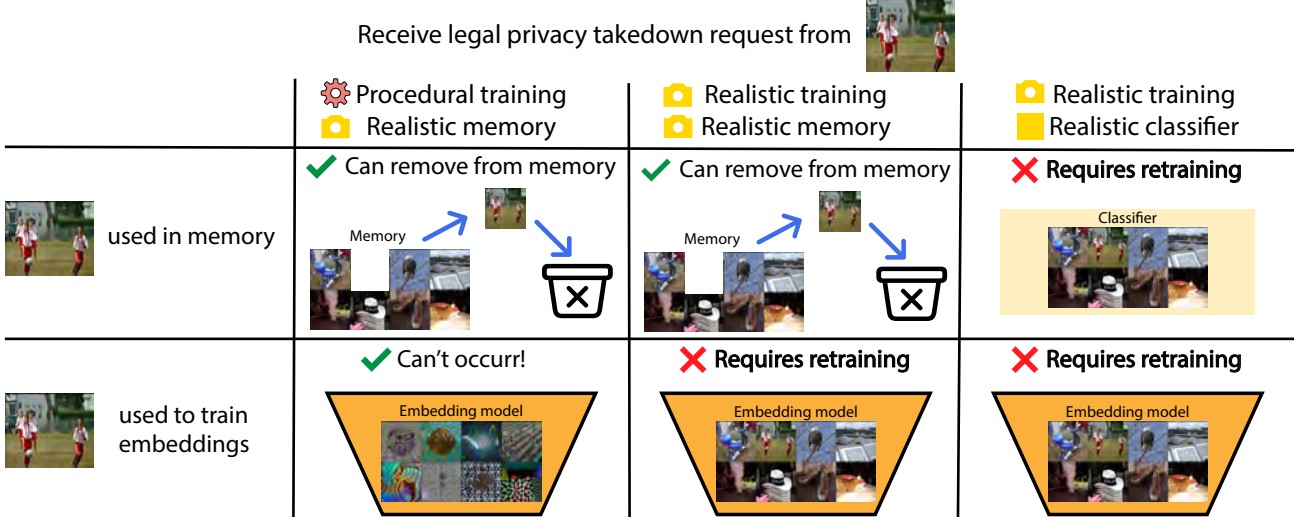

**Figure 13:** Unlearning procedure in response to a legal privacy takedown request. Prior visual memory approaches can efficiently unlearn data in the memory, but not data used to train the embedding model. Our proposal to use procedural data makes the latter case much less likely, as procedural data has drastically lower exposure to privacy risk.

## 6. Efficient unlearning and privacy guarantees

Lastly, we explore how our approach can efficiently unlearn data and compute privacy guarantees, two problems that are expensive and difficult for prior approaches.

**Data unlearning** is the problem of eliminating a piece of data from the weights of a model. This is an important issue especially when it comes to images of humans, NSFW, or illegal content, and has received widespread attention in generative models. Prior work has focused on editing the weights to suppress the concept, but current methods are not infallible and still use the "contaminated" weights as an initial point, which wouldn't satisfy a legal request. Visual memory (Geirhos et al., 2024) offers a compelling solution: simply remove the offending data from the memory. However, this approach fails when the offending data was used to train the embeddings, as seen in Figure 13. Training on procedural data solves this problem elegantly by suppressing this possibility, as it has little privacy risk.

**Differential privacy** characterizes anonymity of individual samples in a training set (Dwork & Roth, 2013; Dwork et al., 2006; Dwork, 2011). For a deterministic model, the definition is: let $A$ be the learning algorithm, $D$ the training set, and $\mathcal{A}(D)$ the model predictions. Then $\mathcal{A}$ has differential privacy for $x \in D$ if $\mathcal{A}(D - \{x\}) = \mathcal{A}(D)$. This is expensive to compute when $x$ was used to train the embeddings: the latter change, so re-training is required. In contrast, for procedural embeddings with visual memory we can simply compare test set predictions with and without any real image $x$ in the memory, which takes little time to compute. Figure 14 plots KNN classification accuracy vs the fraction of non-private training images (those which,

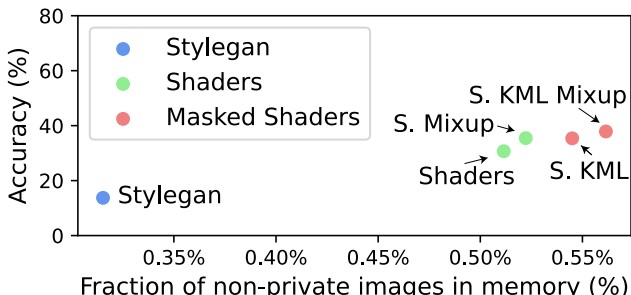

**Figure 14:** Non-private training samples (%) versus KNN classification accuracy on ImageNet for the procedurally trained models. A linear relationship between accuracy and privacy is observed, though only 0.6% of training samples are non-private.

when removed from the memory, change the prediction of at least one test image) on ImageNet-1K. It shows a linear trend between performance and privacy, although the models are quite private as less than 0.6% of the samples are non-private.

**Sensitive data** is information that legally or ethically needs to be handled with high standards of care and control, such as facial identity or medical data. In this scenario, directly training on the data is often not acceptable; procedural models with memory thus offer an elegant solution. Figure 15 shows CelebA (Liu et al., 2015) query images and nearest neighbours according to the S. KML Mixup model. The latter effectively matches appearance and facial expressions despite never training on faces. Table 3 shows classification accuracy on the MedMNIST datasets (Yang et al., 2021), where we see that the procedural models match or exceed the best result from the paper in 7 out of the 10 datasets studied, and obtain good performance otherwise.

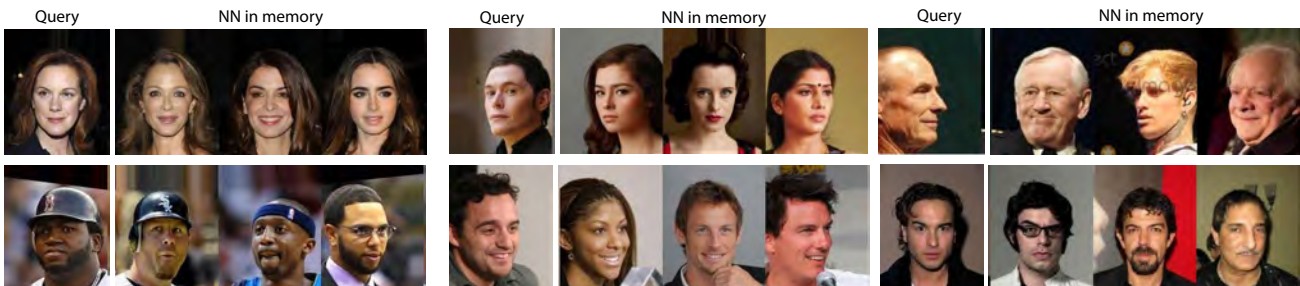

**Figure 15:** Despite never training on faces, the Shaders KML Mixup model can match for appearance and facial expressions on CelebA.

| Data type | Dataset | Path | Derma | OCT | Pneum | Breast | Blood | Tissue | OrganA | OrganC | OrganS |
|---|---|---|---|---|---|---|---|---|---|---|---|
| Best from original paper | | 91.1 | 76.8 | 77.6 | 94.6 | 86.3 | 96.6 | **70.3** | **95.1** | 92.0 | 81.3 |
| Realistic | ImageNet | 98.34 | 82.05 | **95.11** | 95.23 | 91.03 | 96.79 | 57.00 | **94.33** | **93.81** | **85.89** |
| | Places | 98.21 | 80.96 | 94.52 | 96.18 | 91.03 | 95.85 | 54.83 | 90.09 | 92.56 | 82.26 |
| Procedural | S. KML Mixup | 98.60 | 81.36 | 92.85 | **96.56** | **93.59** | 95.09 | 55.62 | 83.76 | 81.48 | 74.43 |
| | S. KML | **99.30** | **82.15** | 92.12 | **96.56** | 91.03 | 96.26 | 58.02 | 85.07 | 85.37 | 81.57 |
| | S. Mixup | 98.88 | 81.56 | 89.14 | **96.56** | 92.31 | 96.73 | 53.27 | 81.96 | 82.53 | 74.10 |
| | Shaders | 98.96 | 81.26 | 92.09 | 95.61 | 92.31 | **97.72** | **58.74** | 86.83 | 86.04 | 81.69 |
| | Stylegan | 98.05 | 77.67 | 85.04 | 95.04 | 87.18 | 91.59 | 53.10 | 76.81 | 77.30 | 76.06 |
| White-box | Random init. | 73.05 | 67.50 | 66.39 | 84.54 | 83.33 | 85.51 | 44.27 | 75.07 | 68.94 | 60.93 |

**Table 3:** KNN classification accuracy on the MedMNIST datasets. Procedural models match or exceed the best result from the original paper (Yang et al., 2022) (a normally trained ResNet) in 7/10 datasets.

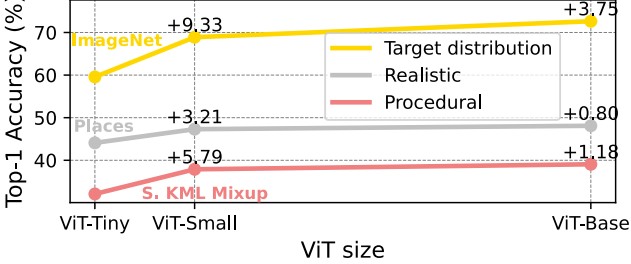

**Figure 16:** KNN classification accuracy on ImageNet-1K as Vision Transformer (ViT) size is scaled. Procedural models dont overfit as size increases; higher capacity yields higher performance.

## 7. Model size scaling

Figure 16 plots classification accuracy on ImageNet-1K versus model size. We observe that models do not overfit to procedural data as capacity increases, suggesting that with larger models performance increases.

## 8. Storage, computation, and accuracy trade-offs of memory-based versus parametric classifier-based approaches in practical deployments

Choosing between a memory-based approach and a parametric classifier-based parametric approach in practice involves a number of trade-offs.

**Training cost:** First and foremost is the computational cost of training. Training the linear classifier required 4 GPU-days on an 8-V100 node, while computing the embeddings of KNN classification required just 1.5 GPU-hours and is doable on a single GPU. The ratio is ∼64X. If the set of classes (the "world") is static, then the initial training is a one-time cost that may be acceptable. However, the real world is anything but static, and so re-training costs can quickly spiral.

**Inference cost:** On the original research code, the inference costs of the KNN classifier were about double of the linear classifier's (1min52s vs 4min for the entire ImageNet validation set of 50k). However, dedicated efficient nearest neighbour search libraries such as faiss (Douze et al., 2024; Johnson et al., 2019) can make search much more efficient in a production setting i.e. with queries within a 10M database taking <0.03ms.

**Memory cost:** The storage requirements is where the memory-based approach is most demanding in comparison with the classifier approach, as the former scales with the number of examples while the latter with the number of classes. On ImageNet, this ratio is about 1000X. However, modern storage technology is incredibly cheap especially compared to GPUs. Storing the entire ImageNet embeddings (384 floats x 1.3M examples) would require ∼2GiB, and 1000GiB SSDs may be purchased online for ∼100USD. In contrast for GPUs, a single V100 has MSRP around

10,000USD while a single H100 has MSRP 30,000 USD, which are needed to train the models. Essentially, storage costs are almost negligible compared to training and inference costs.

## 9. Conclusion

Deep learning is an extremely powerful framework, capable of learning with very little explicit structure by optimizing weights on data. However, representing knowledge via weights has some important drawbacks. In particular, it is very difficult to add, remove, and edit knowledge once its been put into weights. As models scale, retraining or fine-tuning becomes more and more costly, especially in a future where data is licensed instead of bought, as most digital goods are right now. Correcting mistakes or eliminating outdated data as our understanding of reality grows is also of high interest. Privacy-wise, regulations currently limit many AI tools in the EU. To this end, prior work proposed an explicit separation between how knowledge is represented, the feature embeddings, and what knowledge is stored, the visual memory. Taking the form a database of instances, editing knowledge in the visual memory is as simple as adding or dropping data. However, a problem remained: the feature embeddings themselves were generated through weights trained on real-world data, where unlearning remained difficult. In this work, we proposed training the embeddings with procedural data. Unlike real-world data, procedural data is non-realistic and generated via simple code, and thus is much less exposed to the privacy or bias risks that necessitate unlearning. Combining procedural embeddings and visual memory results in a system where *all* real world data can be flexibly added, removed, and provably evaluated for privacy, while approximating the performance of standard methods.

## Acknowledgements

This work was supported by the Defense Science and Technology Agency, Singapore. Additionally, Adrian Rodriguez-Muñoz was supported by the LaCaixa fellowship (LCF/BQ/EU22/11930084).

## Impact Statement

Our work contributes towards AI models that are more transparent, controllable, and applicable in domains with strict data regulations. Moreover, minimizing fine-tuning or retraining significantly reduces power demands and allows remaining power to be allocated in more useful ways.

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

# A. Training details

We trained a vision transformers (Small ViT) (Dosovitskiy et al., 2021) for each dataset (ImageNet, Places, Shaders KML Mixup, Shaders KML, Shaders Mixup, Shaders, and Stylegan), using the recipe and architecture of the original DINO paper (Caron et al., 2021). In particular, we used the optimal hyperparameters for the model trained on ImageNet for all models, rather than hyper-optimizing for performance on each specific dataset. This results in a much more rigorous evaluation, as the optimal ImageNet hyperparameters are more likely to be bad than good for procedural non-realistic data. These hyperparameters are: learning rate 1e-3, batch size 512, optimizer AdamW, num epochs 100, and DINO head out dim 65536. All models are then used without any fine-tuning to obtain all the results, including Figure 5, Table 1, Figures 9 and 10, and Tables 2 and 3.

# B. Additional results

## B.1. Benchmark saturation on NIGHTS

In Figure 5 it visually appears that Shaders-based procedural models are all quite close in performance to each other and to the realistic Places model. To quantitatively test this, we performed a z-test and determined that Places, S. KML, and Shaders are all equivalent to the 5% level. This supports the finding that procedural models have reached the level of real models this benchmark. For the z-test, we used the average NIGHTS results and number of samples in the val dataset (1720). We also include standard deviations of the mean for reference in Table 4.

| Dataset Type | Dataset | NIGHTS-Val |
|---|---|---|
| Target | ImageNet | $0.8733 \pm 0.0080$ |
| Realistic | Places | $0.8331 \pm 0.0090$ |
| Procedural | S. KML Mixup | $0.8105 \pm 0.0095$ |
| | S. KML | $0.8244 \pm 0.0092$ |
| | S. Mixup | $0.8110 \pm 0.0094$ |
| | Shaders | $0.8169 \pm 0.0093$ |
| | Stylegan | $0.7605 \pm 0.0103$ |

**Table 4:** NIGHTS-Val performance with standard deviations.

## B.2. Linear decoding performance

| Dataset Type | Dataset | KNN | Linear |
|---|---|---|---|
| Target | ImageNet | 68.9 | 73.6 |
| Realistic | Places | 47.3 | 59.6 |
| Procedural | S. KML Mixup | **37.9** | **47.3** |
| | S. KML | 35.4 | 47.1 |
| | S. Mixup | 35.4 | 44.8 |
| | Shaders | 30.7 | 43.1 |
| | Stylegan | 13.7 | 26.4 |

**Table 5:** Top-1 accuracy results for linear decoding on ImageNet-1K. With linear decoding, S. KML beats S. Mixup by 2.2%, despite the KNN performances being equivalent. Moreover, the gains from adding Mixup to both Shaders and S. KML are much smaller. This suggests that Mixup mainly reduces bad features, which can also be pruned by the decoder, while KML yields either better or a greater amount of useful features. The two approaches are complementary, which is why Shaders KML Mixup obtains the strongest performance overall.

## B.3. Linear decoding GradCam

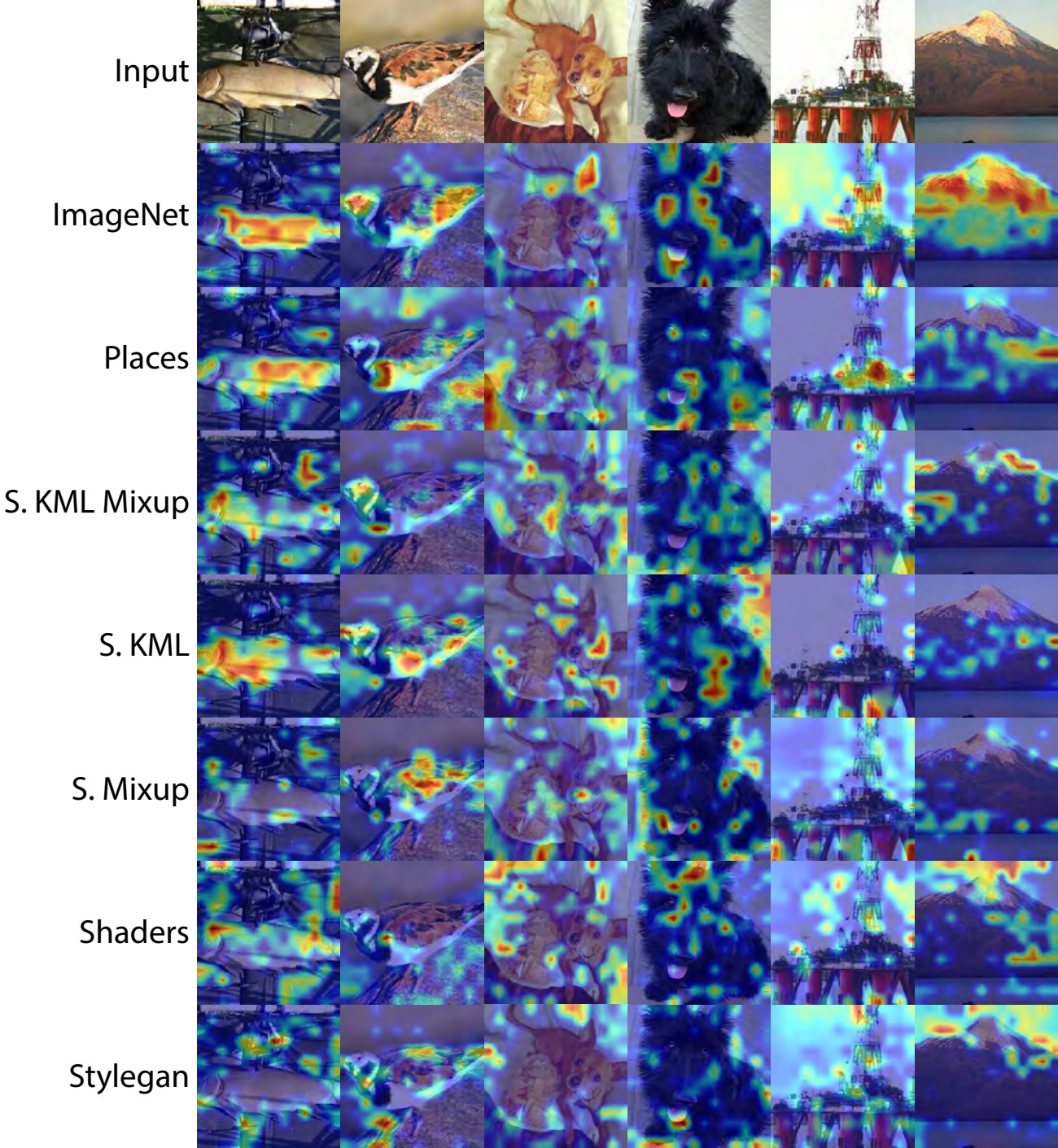

**Figure 17:** GradCam (Selvaraju et al., 2019; Gildenblat & contributors, 2021) visualization for linear decoding on random images from ImageNet for each of the models.

## B.4. Hard-label unsupervised segmentation with K-Means

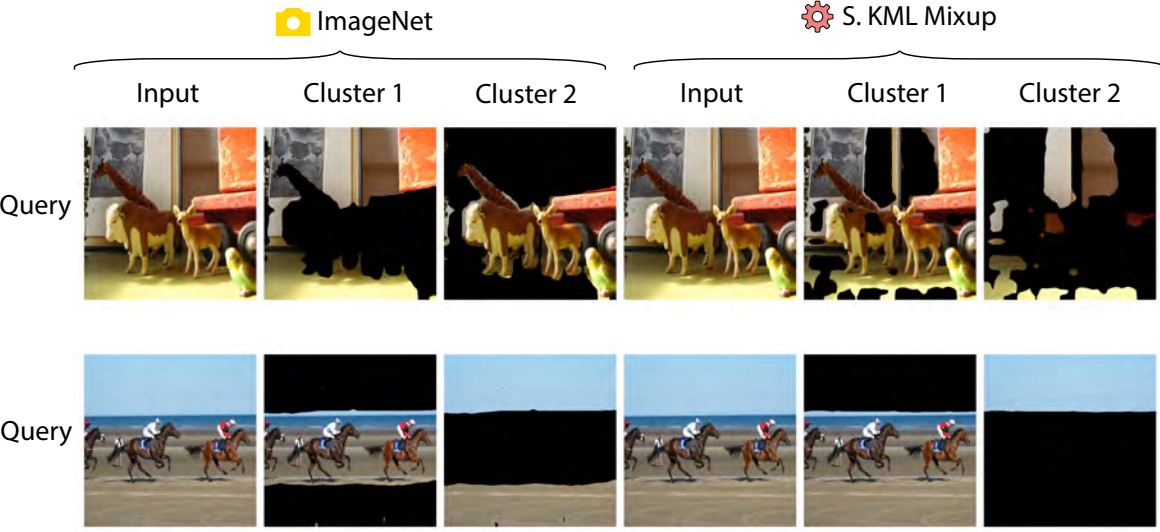

**Figure 18:** Hard-level unsupervised segmentation with K-Means on COCO. The procedural S. KML Mixup model makes visually sensible clusters, but that do not reflect real-world objects, unlike the ImageNet model. This results in incorrect class nearest neighbours, as seen in Figure 19

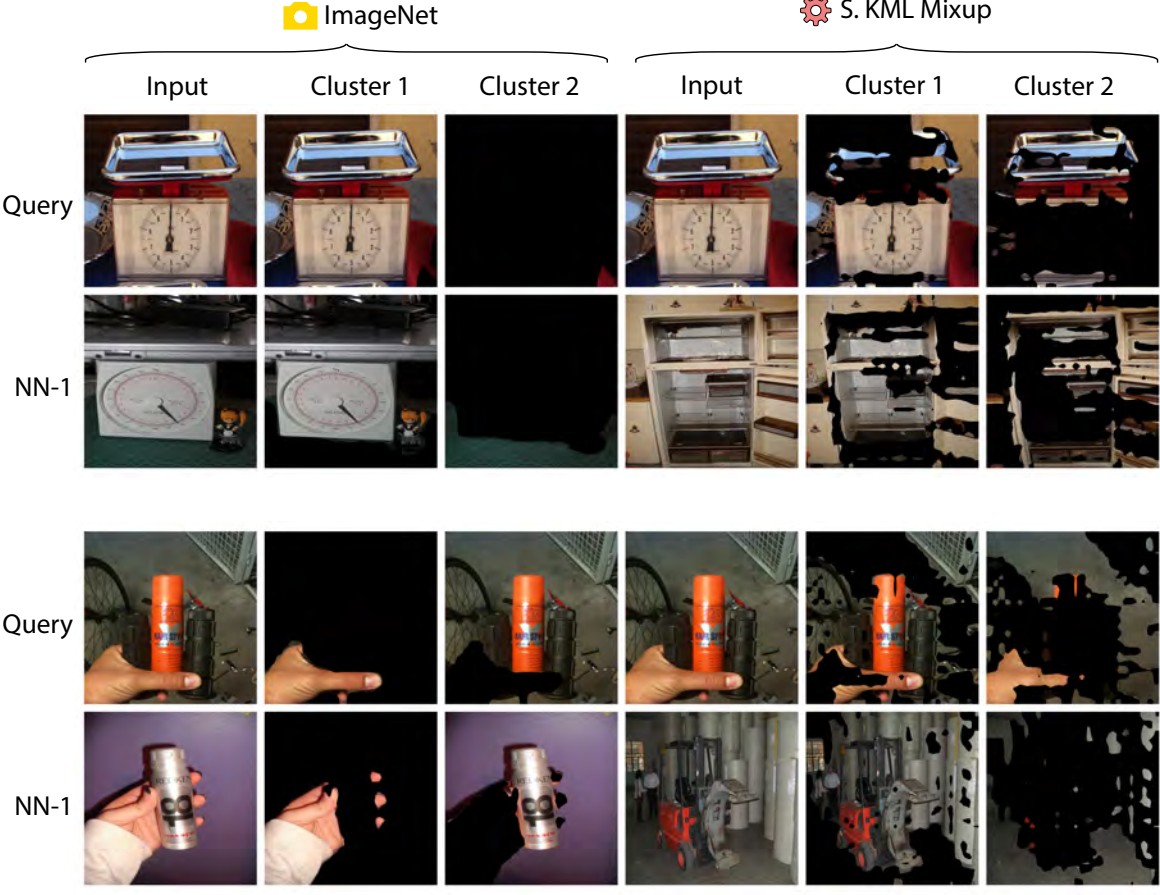

**Figure 19:** Hard-level unsupervised segmentation with K-Means on ImageNet of a query image and is Nearest Neighbour (NN-1). The procedural S. KML Mixup model makes visually sensible clusters, but that do not reflect real-world objects, unlike the ImageNet model. The clusters of the query and NN-1 images visually resemble each other, but since the visual resemblance is not aligned with real world objects, the choices are incorrect for classification.

