# OpenReview forum: "Separating Knowledge and Perception with Procedural Data"
_ICML.cc/2025/Conference — ICML 2025 poster_

### Official Review · Reviewer_ZRET · 2025-03-12

**Overall Recommendation:** 3

**Summary:**

The paper introduces a novel method to fully compartmentalize visual memory by training representation models exclusively with procedural data, thus eliminating the risks associated with privacy and bias inherent in real-world data. The main findings include achieving near state-of-the-art performance on standard benchmarks: procedural models perform comparably to or better than models trained on real data on fine-grained classification tasks and show strong zero-shot segmentation abilities. Importantly, the approach enables perfect data unlearning by simply removing images from the visual memory database, without requiring retraining.

**Claims And Evidence:**

The paper's claims are well-supported by experimental evidence.

**Essential References Not Discussed:**

The paper cites key related work effectively.

**Experimental Designs Or Analyses:**

The experimental analyses are generally sound. Specific analyses reviewed include:

1. Classification accuracy across multiple fine-grained and general datasets: Methodology sound; experiments clearly conducted. But the details need to be clarified.
2. PCA and visual analysis of representations: Provides a helpful and rigorous visual assessment of model embedding qualities and limitations.
3. Privacy evaluation: The comparison of memory-based model accuracy vs. non-private training images is insightful and convincing.

**Methods And Evaluation Criteria:**

The methods and evaluation criteria are well-chosen and appropriate.

**Other Comments Or Suggestions:**

See questions.

**Other Strengths And Weaknesses:**

Strengths:

1.  The approach is novel, clearly motivated, and practical in contexts with high privacy concerns.
2.  Experiments and analyses are thorough, demonstrating effectiveness in multiple domains, including fine-grained classification and segmentation.
3. Well-organized and clearly written, with useful visualizations (e.g., Figures 9 and 10).

Weaknesses:

1.  Limitations around segmentation due to excessively local procedural embeddings are noted but not deeply addressed. Further insights or solutions could strengthen the paper.

**Questions For Authors:**

Can you clarify all baselines' settings? For example, how to train and fine tune the models in Table 1? What dataset is used?

**Relation To Broader Scientific Literature:**

The paper situates itself well within existing literature on procedural data learning, memory-based models, and privacy in AI. It builds explicitly upon previous procedural image generation work and visual memory approaches. The comparison with prior methods is clearly articulated, and contributions such as new procedural data generation methods are well-positioned within the existing literature.

**Theoretical Claims:**

The paper makes no explicit formal theoretical claims that require proof verification.

---

> ### Author Rebuttal · Authors · 2025-04-01
>
> We thank the reviewer for their comments and suggestions, and for agreeing that our approach is novel and shows strong abilities with perfect data unlearning. Below, we clarify how to run our baselines. We will also add a new figure to the camera ready that makes the connection between the limitations at KNN segmentation and classification more explicit.
>
> ## 1 Clarification of baselines
> We train a vision transformer (Small ViT) for each dataset (ImageNet, Places, Shaders KML Mixup, Shaders KML, Shaders Mixup, Shaders, and Stylegan), using the recipe and architecture of the original DINO paper [1]. In particular, we used the hyperparameters that yielded the best results on the original DINO on ImageNet for all models, rather than hyper-optimizing for performance on each specific dataset. This results in a much more rigorous evaluation, as the optimal ImageNet hyperparameters are more likely to be bad than good for procedural non-realistic data. These hyperparameters are: learning rate 1e-3, batch size 512, optimizer AdamW, num epochs 100, and DINO head out dim 65536. We will include them in the Supplementary Material of the camera ready.
>
> These models are then used without any fine-tuning to obtain all the results, including Figure 5, Table 1, Figures 9 and 10, and Tables 2 and 3.
>
> ### 1.1 Table 1
> As mentioned above, we train a single ViT model on each real and procedural dataset. The second column of Table 1 shows the dataset, while the first shows the type of data: target, realistic, and procedural. The white-box models are non-neural approaches included for reference. We obtain the numbers using the nearest networks evaluation method of the original DINO paper [1]. Normalized embeddings are calculated for the train and validation splits, and class predictions for the validation examples are obtained by taking a majority vote of the nearest neighbours in the training set. We evaluate all models using 10, 20, 100, 200 neighbours and keeping the best result.
>
> ### 1.2 Table 2 and Figure 9
> For Table 2, we again take the exact same models (without further training) and compute dense embeddings on the COCO validation dataset. We then apply PCA with 3 channels, which are visualized in Figure 9, and compute their R^2 correlation with the human segmentation labels.
>
> ### 1.3 Table 3
> For Table 3, we do the same as for Table 1 but with the medical MNIST datasets.
>
> ### 1.4 Figure 5
> For Figure 5, we use the original dreamsim [2] code for evaluating baseline models. In particular, they obtain embeddings for the reference, option A, and option B images, and choose option A if it has greater cosine similarity with the reference than B and vice-versa.
>
> ## 2 Expanding on excessively local embeddings
> We will include the hard-label equivalents of Figure 9, which the reviewer thought was insightful, and Figure 12 into the camera ready. In contrast to the soft-label PCA, this figure shows a hard segmentation of the image. The soft-label PCA provides an overview of the model’s internal representation at all granularities, while the hard-label PCA delves deeper into a specific granularity. Combined they offer a more complete analysis of the model’s capabilities.
>
> From the hard-label PCA, we gain further evidence that while realistic data models recognize and segment images into objects, procedural data models cannot due to having never seen them during training. This figure makes the link between limitations at KNN segmentation and classification explicit. Excessively local representations that do not encompass whole objects impair searching for similar images both locally for segmentation, and globally for classification (when pooled into a class token).
>
> A potential solution could consist of more sophisticated algorithms than Nearest Neighbours that take into account a larger context and thus can bypass the excessive locality limitation, but we leave this to future work.
>
> If the reviewer found these additional experiments and explanations convincing, we kindly ask them to raise their score to Accept.
>
> - [1] Emerging Properties in Self-Supervised Vision Transformers; Caron et al.
> - [2] DreamSim: Learning New Dimensions of Human Visual Similarity using Synthetic Data; Fu et al.

---

### Official Review · Reviewer_YM9V · 2025-03-14

**Overall Recommendation:** 2

**Summary:**

This paper introduces a memory-based approach to visual perception by training an embedding model solely on procedurally generated data, then using real data embeddings in a separate memory for classification and segmentation tasks. The authors emphasize advantages in unlearning and privacy, aiming to decouple training on real data from downstream usage.

**Claims And Evidence:**

The authors claim improved control over unlearning and privacy with minimal reliance on real data, and they provide empirical demonstrations supporting these benefits. Evidence for the overall performance is present but could be strengthened with additional baselines.

**Essential References Not Discussed:**

Related works are well discussed.

**Experimental Designs Or Analyses:**

The unlearning experiments, where samples can be removed from the memory without retraining the embedding model, fit logically with the proposed method. Nevertheless, a direct comparison to standard classifier-based unlearning methods would help clarify whether the current approach strikes the best balance between accuracy and privacy.

**Methods And Evaluation Criteria:**

Using a memory-based k-nearest neighbors (KNN) approach and various procedural data sources makes sense for testing the proposed unlearning/privacy framework. The paper employs established benchmark tasks for classification and segmentation to illustrate feasibility.

**Other Comments Or Suggestions:**

I suggest the authors to additionally evaluate on OOD variants of ImageNet like ImageNet-v2, ImageNet-Sketch, etc.

**Other Strengths And Weaknesses:**

1. Motivation for memory-based approach: Beyond simplifying unlearning, it is unclear what additional advantages this framework offers; the paper’s rationale would benefit from a more convincing demonstration of its utility beyond privacy concerns.

2. Accuracy trade-offs: The KNN-based classification and segmentation yield relatively low accuracy compared to contemporary classifiers (often exceeding 80% on ImageNet-1K). The paper should compare a “classifier + unlearning” pipeline to demonstrate whether the privacy gains truly justify the performance costs.

3. Procedural data domain gap: The reliance on procedurally generated imagery raises questions about domain shifts, as these synthetic images differ stylistically from real-world data. The authors do not fully analyze how this gap may contribute to reduced accuracy, nor provide ablation studies to quantify its impact.

**Questions For Authors:**

See comments above.

**Relation To Broader Scientific Literature:**

This work aligns with existing research on using synthetic data for privacy-preserving machine learning approaches

**Theoretical Claims:**

No formal proofs are offered. The core theoretical motivation revolves around the idea that separating real data from the training process (by relying on procedural data and a memory-based scheme) should mitigate privacy risks. However, the paper does not provide a fully rigorous explanation of how procedural data itself, beyond simpler domain adaptation arguments, translates into enhanced interpretability or privacy guarantees.

---

> ### Author Rebuttal · Authors · 2025-04-01
>
> We thank the reviewer for their insightful comments and for agreeing we provide empirical demonstrations on the benefits of our approach. In particular, we appreciate their identification of areas which could be strengthened, such as clarifying the benefits of our approach and additional baselines, which we write below
>
> ## 1 Motivation for memory approach
> Our work builds on top of prior research on memory, in particular Geirhos et al. 2024. They argue that precisely because the real world changes; a static model (such as a standard classifier) would require constant retraining or fine-tuning, which is not feasible especially given the scale of modern models. A flexible visual memory allows for efficiently adding even billion-scale data, removing data through unlearning, and an interpretable decision-mechanism.
>
> However, prior approaches had a problem: the feature embeddings are themselves trained on data. We identify four consequences. First, while adding and removing data from the memory is easy, doing so from the embeddings is not. Second, a large corpus of data is required to train the embeddings, which could be not available (low-resource setting) or not allowed (privacy setting). Lastly, counterfactual interpretability with data used to train the embeddings is difficult. Our proposal to use procedural data effectively addresses all four.
>
> ### 1.1 Efficient unlearning
> Procedural embeddings allow for efficient provable unlearning of all real data used. Classifier unlearning methods fine-tune or retrain the weights, which is expensive and not infallible. Moreover, they also use the ”contaminated” weights as an initial point which wouldn't satisfy a legal request. In these situations where classifier unlearning is not feasible or allowed, procedural data is effective and efficient
>
> ### 1.2 Low resource setting
> Training an embedding model requires lots of data. In situations where it is not available, procedural data yields strong performance. As seen in Table 1, procedural models actually beat the Places model on some fine-grained classification tasks. This shows that procedural data may sometimes perform better than out-of-distribution real data
>
> ### 1.3 Privacy setting
> If directly training on the data is not acceptable, our approach offers an elegant solution. With procedural embeddings, any medical information is non-existent in the model's weights. Moreover, as we see in Table 3, procedural models match or exceed the best result from the original MedMNIST paper [1] in 7/10 datasets
>
> ### 1.4 Counterfactual interpretability
> Prior work in AI ethics identified the concept of a “right to explanation” to people affected by automated decisions. The paper [2] showed that a “counterfactual”, defined as the data that, if it were absent, would change the model decision, satisfied key requirements for both people and firms. Computation of counterfactuals requires efficient computation of “what if” scenarios in absence of certain data, which we can obtain with unlearning. Our method enables counterfactual analysis for all real data used. Given a decision, we search for the counterfactual by sequentially unlearning the top neighbour and checking if the answer changes
>
> ## 2 Accuracy trade-off
> Our setting is self-supervised learning without labeled data. We use classification only as a proxy task for evaluation. In this setting, training on ImageNet obtains 69% accuracy, while training on realistic data (Places) obtains 47%. Training on procedural data is within 10% of the latter, which is quite significant given it is non-realistic. We are not saying that procedural data should be used when standard classifiers are possible and appropriate, only that in situations like described in point 1, procedural data is highly performant and desirable
>
> ## 3 Procedural data domain gap + additional baselines
> We actually have that discussion in the paper from line 271 to the end of page 5, in Figure 11, and in Section 5. We mention how lack of real-world objects in procedural data leads to the performance gap. The additional baselines requested can be found below. The results are very similar to those in the original paper. On the three ImageNet-v2 variants (INv2 Top Quality, Threshold0.7, and MatchedFrequency), the gap of the best procedural model to Places is still 10%. On ImageNet-Sketch (IN-S) it's even better, as the gap to Places is 3%: procedural models are able to search for similar neighbour sketches. In the fine-grained dataset Stanford Dogs, procedural models again beat Places
>
> | Dataset | INv2-TQ | INv2-Th | INv2-MF | IN-S | Dogs
> | - | - | - | - | - | -
> | ImageNet | 69.9 | 64.4 | 56.6 | 66.5 | 65.6
> | Places | 46.1 | 40.8 | 35.4 | 56.6 | 22.2
> | S. KML Mixup | 35.3 | 30.7 | 26.0 | 52.9 | 29.5
>
> - [1] MedMNIST Classification Decathlon; Yang et al
> - [2] Counterfactual Explanations without Opening the Black Box: Automated Decisions and the GDPR; Wachter et al

---

> > ### Comment · Reviewer_YM9V · 2025-04-09
> >
> > Thanks for the detailed response of the authors. While some of my questions are addressed, my major concern remains unressolved.
> >
> > Compared to merely explaining the advantages of memory-based methods, I would prefer to see direct metrics that prove the superiority of your approach. While the current claims intuitively make sense, they are not sufficiently persuasive. When new data arrives, changing already trained embeddings is indeed more challenging, but it requires less storage space and introduces less additional computational load than KNN. Therefore, the authors need to carefully compare the accuracy trade-offs between your model and traditional classifiers, but I have not seen any explicit analysis and results so far. Moreover, the main work of this paper is applying existing memory-based approaches to procedural data, with limited innovation or contribution to support an ICML-level publication. Thus, I maintain my original score of a rejection.

---

### Official Review · Reviewer_VmuU · 2025-03-17

**Overall Recommendation:** 3

**Summary:**

This paper introduces a novel process for training neural networks using procedural data: Shaders KML and Shaders KML Mixup. Despite relying on simple programs and lacking any direct resemblance to the target distribution, the proposed method achieves impressive results in K-NN-based classification. The paper’s core contribution is K-Means Leaves (KML), a data-driven masking strategy that replaces simplistic masking with k-means clustering to diversify training patterns. This approach significantly enhances the variety of features leveraged during training, addressing a key limitation of traditional procedural data generation. The paper  further demonstrate the versatility of KML by showcasing its benefits across multiple applications, including memory-based classification (via nearest neighbors), semantic segmentation, unlearning tasks, and privacy-preserving scenarios with formal guarantees.

**Claims And Evidence:**

The paper compellingly demonstrates the broad utility of the procedural method for classification and segmentation tasks. That said, the performance improvements between Shaders and Shaders Mixup appear modest in most benchmarks (excluding CUB). This suggests the added value of Mixup in this framework may merit further discussion, particularly regarding the claim that increased dataset diversity is a “key driver of performance.” To bolster this argument, it might help to clarify how the KML masking procedure contributes—for example, by highlighting whether specific components (e.g., feature types, training phases) benefit disproportionately, or if certain induced biases (e.g., geometric consistency) play an underappreciated role.

**Essential References Not Discussed:**

I think the paper cites the most recent work on this domain.

**Experimental Designs Or Analyses:**

I checked the designs for :

* Alignment with humans in the NIGHTS dataset, it would be good to provide significance values on the difference between shaders and the improvement proposed in the paper.
* K-NN classification, in fine grained and general Imagenet datasets.
* Segmentation performance

**Methods And Evaluation Criteria:**

Yes.

**Other Comments Or Suggestions:**

* Perhaps is better to list the method as "ours" across to ease readability and clarity.

**Other Strengths And Weaknesses:**

Strenghts

* The paper shows the application of procedural training on multiple tasks, extending previous papers on the same topic.

* The method is actually quite simple, but it shows promise in improving the frontier on the topic.

Weakness
* It would strength the paper if more evidence on the benefit or the rational before the masking procedure with respect to previous works in the area.

**Questions For Authors:**

* Maybe not super necessary given the approach of visual memory perception, but what is the expected upper bound of these models? i.e.   Linear decoding for classification instead of KNNs
* If the one above is done, an explainability analysis would give also very interesting insights of what features are learned by these models.

**Relation To Broader Scientific Literature:**

The paper contributes to the area of training with procedural data.

**Theoretical Claims:**

No theoretical proofs were considered.

---

> ### Author Rebuttal · Authors · 2025-04-01
>
> We thank the reviewer for agreeing that the approach is novel and has broad utility, as well as their insightful comments, questions, and suggestions. We answer the questions and follow up on the suggestions below
>
> ## 1 Benefits of KML over Mixup
> We agree the benefits of KML may initially appear modest (sans CUB). As suggested, we clarify that KML especially improves the variety of visual stimuli the model is exposed to, which leads to higher variety features for downstream tasks. To show this, we compute the Vendi Diversity (VD) with the ImageNet model and the linear classifier performance. These two metrics provide additional evidence of the benefits of KML complementary to KNN classification accuracy. Moreover, the improvements in KNN classification are better observed by looking at the relative reduction in the gap to real data rather than the raw increase.
>
> ### 1.1 VD
> VD [1] measures the entropy of the eigenvalues of the similarity matrix of pairs in a dataset. It has been used for generative modelling and dataset curation in many domains such as molecules and images. We use the ImageNet model to create the similarity matrix, acting as a proxy for human vision to measure the diversity of visual stimuli in our procedural datasets.
> - S. KML Mixup 38.6
> - S. KML 36.5
> - S. Mixup 30.6
> - Shaders 22.1
>
> Indeed, while Mixup improves the diversity of Shaders by 8.5, our KML improves it by 14.3. Combining KML and Mixup yields a larger improvement of 16.5. Our intent in line 176 was not to make the claim ourselves but rather to cite [2], which showed that improved diversity is a key driver of performance for non-realistic data, as our motivation for developing KML. We apologize for the confusion
>
> ## 1.2 Linear decoding
> We thank the reviewer for their suggestion of analyzing linear decoding. We had actually trained linear decoders on ImageNet, but did not include them initially as we focused on visual memory perception. We will include these numbers, as well as GradCam [3] visualisations for explainability, in the camera-ready supplementary material
> - S. KML Mixup 47.3
> - S. KML 47.1
> - S. Mixup 44.8
> - Shaders 43.1
>
> With linear decoding, S. KML beats S. Mixup by 2.2%, despite the KNN performances being equivalent. Moreover, the gains from adding Mixup to both Shaders and S. KML are much smaller. This suggests that Mixup mainly reduces bad features, which can also be pruned by the decoder, while KML yields either better or a greater amount of useful features. The two approaches are complementary, which is why Shaders KML Mixup obtains the strongest performance overall.
>
> ### 1.3 Relative gap reduction
> Additionally, gains in classification accuracy are better observed by looking at the relative reduction in the gap to real data rather than the raw increase. This is because of diminishing returns as we get closer to the performance ceiling of procedural data (given by realistic data). Measured relatively, our KML reduces the gap by 18.5% and 20.5% on Flowers and ImageNet respectively. The relative improvement in CUB is actually a very similar 20.4%. The Food dataset is a special case, as S. Mixup matches the real data Places but our S. KML Mixup beats it. We calculate relative increase as follows
>
>       (s_kml_mixup_accuracy - s_mixup_accuracy) / (real_data_ceiling - s_mixup_accuracy)
>
> using the ImageNet model as the ceiling for Flowers and CUB classification, and the Places model for ImageNet classification
>
> ## 2 Significance values of NIGHTS
> We agree that differences are minimal. NIGHTS was included as a task where procedural models have reached the level of real models. A z-test determined Places, S. KML, and Shaders are all equivalent to the 5% level. We will add this result and the standard deviations for Fig. 5 to the camera ready
>
> If the reviewer found these additional experiments and explanations convincing, we kindly ask them to raise their score to Accept.
> - [1] The Vendi Score: A Diversity Evaluation Metric for Machine Learning; Friedman et al
> - [2] Learning to See by Looking at Noise; Baradad et al
> - [3] Grad-CAM: Visual Explanations from Deep Networks via Gradient-based Localization; Selvaraju et al

---

### Decision · Program_Chairs · 2025-05-01

**Decision:**

Accept (poster)

**Comment:**

This paper proposes a representation learning approach that is trained exclusively on procedurally generated data and applied to a range of computer vision tasks via visual memory, without requiring additional training on real data. The method shows strong performance while addressing important privacy concerns by avoiding the use of real-world data. The paper is well-written, well-organized, and presents a compelling direction for privacy-preserving visual learning.

During the review process, reviewers raised questions regarding the motivation behind the method design, the specific role of local procedural embeddings, and the generalization capabilities under domain shift. The authors addressed these concerns effectively in the rebuttal, leading to a more favorable assessment.

Post-rebuttal, reviewer YM9V introduced a new concern regarding the balance between storage requirements, computational efficiency, and model performance. While this is a valid point for ensuring practical deployment, AC agrees with the majority opinion that the paper’s strengths—especially in terms of innovation, performance, and privacy benefits—outweigh its limitations.

AC recommends acceptance and strongly encourages the authors to include an analysis or discussion of the trade-offs between storage, computation, and accuracy in the final version to further strengthen the practical relevance of the work.